# Linking electronic mental healthcare and benefits records in South London: design, procedure and descriptive outcomes

Sharon A M Stevelink [1,2] Ava Phillips,[1] Matthew Broadbent,[3] Andy Boyd,[4] Sarah Dorrington [1,3] Amelia Jewell [3] Ray Leal,[1,2] Ioannis Bakolis,[5,6] Ira Madan,[7,8] Matthew Hotopf,[1,3] Nicola T Fear [2,9] Johnny Downs [1,3]

SAMS and AP are joint first authors.
NTF and JD are joint last authors.

For numbered affiliations see end of article.

**Correspondence to**
Dr Sharon A M Stevelink; sharon.stevelink@kcl.ac.uk

## ABSTRACT

**Objectives** To describe the process and outcomes of a data linkage between electronic secondary mental healthcare records from the South London and Maudsley (SLaM) NHS Foundation Trust with benefits records from the Department for Work and Pensions (DWP). We also describe the mental health and benefit profile of patients who were successfully linked.

**Design** A deterministic linkage of routine records from health and welfare government service providers within a secure environment.

**Setting and participants** Adults aged ≥18 years who were referred to or accessed treatment at SLaM services between January 2007 and June 2019, including those who were treated as part of Improving Access to Psychological Therapies (IAPT) services between January 2008 and June 2019 (n=448 404). Benefits data from the DWP from January 2005 to June 2020.

**Outcome measures** The linkage rate and associated sociodemographic, diagnostic and treatment factors. Recorded primary psychiatric diagnosis based on International Classification of Diseases (ICD)-10 codes and type of benefit receipt.

**Results** A linkage rate of 92.3% was achieved. Women, younger patients and those from ethnic minority groups were less likely to be successfully linked. Patients who had subsequently died, had a recorded primary psychiatric diagnosis, had also engaged with IAPT and had a higher number of historical postcodes available were more likely to be linked. Overall, 83% of patients received benefits at some point between 2005 and 2020. Benefit receipt across the psychiatric diagnosis spectrum was high, over 80% across most ICD-10 codes.

**Conclusions** This data linkage is the first of its kind in the UK demonstrating the use of routinely collected mental health and benefits data. Benefit receipt was high among patients accessing SLaM services and varied by psychiatric diagnosis. Future areas of research are discussed, including exploring the effectiveness of interventions for helping people into work and the impact of benefit reforms.

## INTRODUCTION

In the UK, approximately 1.8 million people face long-term sickness absence of 4 weeks or longer, costing our society £100 billion

## STRENGTH AND LIMITATIONS OF THIS STUDY

⇒ This is a novel data linkage between electronic mental healthcare records and benefits records in the UK.
⇒ A strength of this data linkage is the high linkage rate of 92.3%.
⇒ The sample does not include a comparison group (eg, people who did not access secondary mental healthcare services).
⇒ There is no reliable employment variable within the data stating whether someone is currently in or out of work (except for Universal Credit).
⇒ There is a potential for linkage bias as a result of the method used (ad hoc deterministic fuzzy matching) and having no unique identifier between data sets.

annually.[1] Long-term sickness absence is associated with social exclusion, poor health outcomes and high mortality.[2–4] Each year, over 300 000 people are leaving work due to long-term mental health problems.[5] Mental disorders are one of the most common causes of sickness absence and subsequent long-term occupational disability.[6 7] In 2019/2020, 17.9 million working days were lost due to mental ill health.[8] For many who access mental health services, their difficulties impact on their ability to work. Understanding people's finances, welfare, benefits and occupational needs are integral to the care and quality of life for people with mental disorders; however, these are often overlooked.

Over the last 15 years, major changes have taken place in the UK benefits system including the extension of benefit sanctions[9]; the introduction of 'Universal Credit' (UC), a means-tested benefit replacing six benefits plus tax credits for those of working age[10]; the replacement of personal capability assessments with work capability assessments (WCA) where one's capability for all work-related activity is reviewed; and an increased reliance on conditionality meaning that

people need to fulfil certain work-related activity requirements to maintain their full benefit entitlements. These were announced as part of the Welfare Reform Act 2007 and 2012 and Welfare Reform and Work Act 2016. These changes have been met with concern about their potential impact on people's well-being and particularly on those with mental disorders.[11–16] Hence, research into the welfare and benefit needs of the population with mental disorders is required to inform policy on welfare provision when this group is at their most vulnerable and also to support return to work as an integral part of recovery for people who are able to return to employment.[17 18] The latter is especially relevant given the introduction of, for example, Improving Access to Psychological Therapies (IAPT) services[19] and Individual Placement and Support Services[20] in the UK.

There are no pre-existing datasets in the UK that can currently address this. Alone, National Health Service (NHS) healthcare records are an unreliable source of information on benefit receipts or employment status; these are not routinely collected or recorded. Data held by the Department for Work and Pensions (DWP), which records national welfare and public service interactions, for example, on unemployment-related benefits, lacks high-quality information about health status. The limited data that are available in these benefits records are solely based on diagnostic information provided in benefit applications for specific benefits and these are often incomplete.

The advent of electronic healthcare records and systems, and the increasing sophistication with which data can be linked and analysed, has presented the opportunity to change the academic research landscape. We report here on a unique linkage of welfare and benefits data with routinely collected mental health data of over 400 000 adults referred to psychiatric services, enabling us to address gaps in evidence regarding the interrelationships between benefit receipt, employment status, mental disorders, treatment, well-being and recovery. To our knowledge, this is the first time in the UK that routine health records have been linked with benefits data. However, research into welfare and mental health using data registries have been led by those in Nordic countries where a unique personal identifier is available to all those with a permanent residence record, paving the way for opportunities in linkages between health and welfare registers.[21–23]

Here, we describe the process and outcomes of linking electronic mental healthcare records from patients who accessed secondary mental healthcare services at the South London and Maudsley (SLaM) NHS Foundation Trust with benefits records from the DWP. First, we will describe the ethical and governance considerations encountered before we could proceed with the linkage. Second, we describe the approach, data linkage rate and factors associated with successful linkage. Finally, we provide an overview of the mental health and benefit profile of patients who were successfully linked.

## METHODS
### Data sources
#### SLaM NHS Foundation Trust Biomedical Research Centre (BRC) Case Register

The SLaM NHS Foundation Trust is one of Europe's largest providers of secondary mental healthcare services, providing care predominantly for the South London boroughs of Lambeth, Lewisham, Southwark and Croydon, covering a catchment area of over 1.2 million residents. SLaM provides specialist (secondary) mental healthcare services as well as IAPT services. The SLaM BRC Case Register includes electronic mental healthcare records of patients accessing SLaM. In 2008, the Clinical Records Interactive Search (CRIS) system was developed[24] to curate deidentified data from SLaM's electronic mental healthcare records for research use. Information concerning patients' mental healthcare journey is available in pseudoanonymised format either in free clinical text notes or structured fields as part of a patient's electronic mental healthcare record. CRIS clinical data may include, for example, individual level data on sociodemographic characteristics (eg, month and year of birth, sex, ethnicity, neighbourhood deprivation), time variant data on International Classification of Diseases (ICD)-10 psychiatric diagnosis, diagnostic assessments, mental health treatment (eg, local or specialist services, community vs inpatient), service use (eg, patterns of engagement), medication prescriptions and psychotherapeutic interventions. For the current paper, only data from structured fields were used. CRIS data covered the 1 January 2007 until 30 June 2019.

#### DWP benefits data

The DWP is responsible for the implementation of policy regarding most welfare and state benefits in Great Britain. Benefits data includes individual level demographic data (eg, date of birth, date of death and sex), time variant data related to the on and off flows of benefits (eg, Incapacity Benefit (IB), Carer's Allowance (ICA), Income Support (IS), Housing Benefit (HB), Jobseeker's Allowance (JSA), Attendance Allowance (AA), Retirement/State Pension (RP), Disability Living Allowance (DLA), Severe Disablement Allowance (SDA), Widow's Benefit (WB), Pension Credit (PC), Passported Incapacity Benefit (PIB), Bereavement Benefit (BB), Employment and Support Allowance (ESA), UC, Personal Independence Payment (PIP) and relevant benefit-specific details.[25] Start and end dates of benefit spells are provided as well as the amount of money received. In addition, some information is provided about WCA and work programme participation. Benefits data covered the 1 January 2005 until 30 June 2020.

### Sample

The sample consists of all adults (aged 18 years and older) who (1) have been referred for treatment with

SLaM secondary mental healthcare services between 1 January 2007 (the implementation of electronic mental healthcare records across SLaM secondary mental healthcare services was only finalised by that time) and 30 June 2019 or (2) had an event with SLaM secondary mental healthcare services during this time period and were aged 18 or over at the time of their latest recorded event in the window or (3) had a treatment episode at IAPT between 1 January 2008 and 30 June 2019. Patients ranged in symptom severity from common mental disorders to serious mental illness (eg, schizophrenia, bipolar affective disorder), substance use disorders and organic disorders (eg, neurological syndromes associated with severe intellectual impairment). For the current paper, we only focused on the linkage of patients who accessed specialist (secondary) mental healthcare services within SLaM (and possibly also IAPT) but not those who only accessed IAPT within SLaM. This decision was made as we were especially interested in the former group of patients who were more likely to have severe mental health symptomatology.

## Patient and public involvement and engagement
The proposed linkage of electronic mental healthcare records of SLaM and benefits records from the DWP was presented to the Maudsley Biomedical Research Centre Data Linkage Service User and Carer Advisory Group in December 2016.[26] The members of the Advisory Group experienced mental ill health themselves or as a carer for someone with a mental health diagnosis and were accessing or had accessed mental healthcare services. All were given training concerning data linkages, the underlying clinical research information system, data security, governance and the research environment at SLaM.

The members of the Advisory Group were supportive of the proposed linkage when first discussed in December 2016. The linkage was presented again in September 2019 with a discussion around the specific research questions and opportunities for continued patient and public involvement in the project. They will be consulted on a regular basis now the data linkage has been finalised with a focus on discussing preliminary results and gathering input regarding dissemination and impact strategies.

## Data linkage process
The linkage of CRIS clinical records with benefits data took place in late 2020. An ad hoc deterministic matching approach was used, namely fuzzy matching, based on personal identifiers held on the DWP's Customer Information System, which hosts a 'spine' record of everyone who has ever been issued a National Insurance Number (NINO). The NINO is a unique individual ID allocated for employment, tax and welfare purposes.
1. The SLaM Clinical Data Linkage Service, 'a trusted third party', shared the personal identifiers of the eligible sample (patient name, date of birth, sex, postcode and postcode history) and the BRCID pseudonym

used within the CRIS database with DWP. The data were transferred using the secure 'Egress' system.
2. The DWP linked the SLaM personal identifiers to DWP held personal identifiers in a secure area using a fuzzy-matching process (uniqueness cut-off threshold of 90% or above) to create a table linking the BRCID pseudonym to a NINO (where possible). Approved benefits data were extracted from DWP systems using the NINO.
3. The NINO was replaced with the BRCID pseudonym before the linked deidentified DWP benefits data were sent back to the SLaM Clinical Data Linkage Service via Egress. At no point were SLaM clinical data shared. DWP destroyed the SLaM personal identifiers once the matching work was complete.
4. The benefits data with the attached BRCIDs are stored within the SLaM secure research system in a separate database to the CRIS clinical data with access to restricted users only.
5. The benefits data and CRIS clinical data are only joined on a project-specific basis, after the necessary approvals have been given. BRCIDs are stripped before a project-specific anonymised data set is provided to the researcher.

## Materials
The following sociodemographic and clinical, diagnostic and treatment variables were derived from the linked data for further exploration. These were selected based on data availability, previous research indicating that these factors were found to be associated with data linkage success[27 28] and discussions within the wider research team.

### Sociodemographic variables
All sociodemographic variables were derived from the clinical data, except for patient sex (male/female) as this was more complete in the benefits data. However, if sex was missing in the benefits data, and available in the clinical data, this was backfilled accordingly. Age was calculated using month and year of birth until the SLaM window end date (30 June 2019). Subsequently, age was grouped in the following categories: ≤20, 21–40, 41–60 and >60. Ethnicity was categorised as follows: white/black, African, Caribbean, black British/Asian, Asian British/ Mixed, Multiple racial and ethnic groups/Other racial and ethnic minority groups and 'not stated'. We also had information on whether people had died (month and year) that resulted in a binary death (yes/no) variable. The Index of Multiple Deprivation (IMD) was informed by 2019 data, and we used the postcode closest to and before the SLaM window end date to inform IMD quintiles, with the first quintile indicating most deprived and fifth quintile least deprived. IMD is a summary measure of relative deprivation informed by seven domains, namely income, employment, education, crime, housing, health and living environment at lower levels of geography.[29] We created a variable indicating whether patients lived in the local catchment area based on Lower-layer Super Output

Areas, a small geographical area covering a similar population size, again using the postcode closest to and before the SLaM window end date.[29] In addition, we generated a categorical variable indicating the number of historical postcodes sent to DWP to facilitate the linkage for each patient (up to five maximum).

### Diagnostic and treatment variables

We created a binary primary psychiatric diagnosis variable (yes/no) that referred to whether a psychiatric primary diagnosis was recorded in a patient's record closest and before the SLaM window end date (30 June 2019). This only included the ICD-10 'F codes' referring to mental and behavioural disorders, thereby excluding non-specific diagnoses (eg, Z*, F99*, FXX). Subsequently, we derived a variable outlining the type of diagnosis code patients were given, if any (ranging from F00–F09 (mental and behavioural disorders, and mental disorders due to known physiological conditions) to F90–F98 (behavioural and emotional disorders with onset usually occurring in childhood and adolescence). We also explored whether patients had accessed IAPT (yes/no), in addition to SLaM services between 2008 and 30 June 2019. IAPT was only introduced in 2008 so this was the earliest possible start date. Two binary variables were created (before and after 2013) to indicate patients' first and last contact with SLaM. This cut-off was chosen as PIP was introduced in 2013. Age at first presentation to SLaM (≤20, 21–40, 41–60, >60) was calculated using month and year of birth and the patients' earliest accepted referral date to SLaM closest to and before the SLaM window end date.

### Benefits variables

Participants who were successfully linked to a NINO and had received one of the following benefits between 1 January 2005 and 30 June 2020 were identified as benefit recipients: ESA, JSA, IS, HB, DLA, IB, AA, RP, PIP, UC, PC, ICA, SDA, PIB, BB or WB.[25] We also had information on what UC conditionality regime patients were allocated to namely (1) searching for work, (2) working, with requirements, (3) no work requirements (4) working, no requirements, (5) preparing for work or (6) planning for work.[30]

### Statistical analysis

#### Analysis of linkage bias

All statistical analyses were performed using the statistical package STATA (V.15). All variables were checked for completeness and outliers. Variable completeness and accuracy were improved by backfilling data (using the clinical or benefits records were possible). If outliers were identified, for example, date of birth (as based on the age inclusion criteria), this was recoded as missing (n=14). The same was done for negative values (eg, age at first contact n=192) and improbable dates (eg, having accessed SLaM before it was established n=2210).

The overall linkage rate was determined by calculating the proportion of unique BRCIDs successfully linked to a NINO. We did not expect all patients to have engaged with the DWP to apply for benefits or subsequently successfully received benefits. For example, some participants engaged with the DWP, and a note was made on their benefits record, but they did not meet the criteria to receive, for example, ESA. Therefore, of those successfully linked to a NINO, we also calculated the proportion who had engaged with the DWP, as well as the proportion who had engaged and successfully applied for benefits according to the benefits records.

We then conducted univariable logistic regression analysis to explore sociodemographic, diagnostic and treatment related factors, associated with linkage to benefits records. We also conducted multivariable analyses thereby adjusting for factors identified a priori (namely age, sex and ethnicity).[24 25] Subsequently, we generated a probability estimate of matching as a function of the risk variables with the use of the logistic regression model.

### Sample profile

Multivariable logistic regression models were also employed to explore factors associated with benefit receipt, adjusting for age, sex and ethnicity. In addition, descriptive statistics were used to describe the benefit and the mental health profile of successfully linked patients. The latter was based on the most recently recorded ICD-10 primary psychiatric diagnostic code. We also tabulated the mental health profile of our sample by type of benefit receipt. OR, adjusted OR, 95% CI and p values are reported.

## RESULTS

### Overview of data linkage process and analysis of linkage bias

Unique IDs of 448 404 patients who accessed SLaM services (specialist (secondary) mental healthcare services and/or IAPT) were sent to the DWP (figure 1). For this study, we only report on patients who accessed secondary mental healthcare services at SLaM (n=239 714). Of these, 221 243 (92.3%) were successfully linked to a NINO held by the DWP. Individuals identified as being under the age of 16 according to the personal details held by the DWP and those who resided in Northern Ireland at some point during benefit receipt were excluded from the data sent back to the SLaM Clinical Data Linkage Service, resulting in 220 332 (91.9%) unique linked IDs available for research purposes.

Results from adjusted logistic regression analyses indicated that the following groups of patients were less likely to be linked (an OR greater than 1 denotes greater chance of successful linkage compared with the reference): female patients versus male patients, ethnic minority groups versus patients from a white ethnic background and patients with only one postcode available versus two or more postcodes. Compared with younger patients (<21 years), middle-aged patients (21–60 years) were less likely to be successfully linked, whereas older patients (>60 years) were more likely to be linked compared with

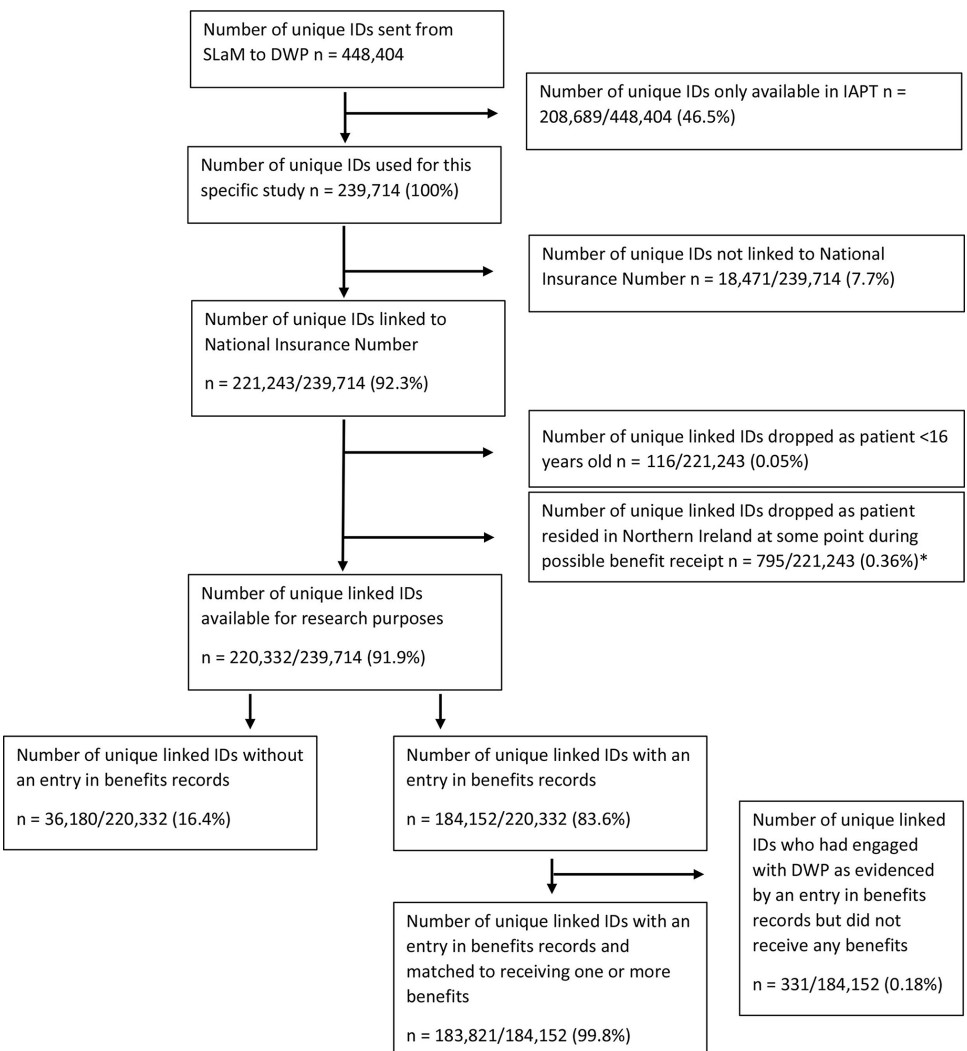

**Figure 1** Overview of BRC patient IDs that were and were not linked to benefits data from the DWP via their National Insurance Number. *The Health Research Authority approval that was received for the data linkage only applies to England and Wales. In addition, the devolved legislature of Northern Ireland is responsible for administering benefits to patients who resided in Northern Ireland at the time of their benefit receipt. Therefore, the DWP do not have authority to share these data. DWP, Department for Work and Pensions; IAPT, Improving Access to Psychological Therapies; SLaM, South London and Maudsley NHS Foundation Trust.

all other age groups (table 1). We also found that those who had died, had a recorded psychiatric primary diagnosis, had engaged with IAPT and accessed SLaM services more recently were more likely to be successfully linked (table 2).

### Sociodemographic-related, diagnostic-related and treatment-related factors associated with benefit receipt

Of the patients who were successfully linked, 184 152 (83.6%) had engaged with the DWP, meaning they had a benefits record but not necessarily successfully claimed benefits. Among the successfully linked patients who had engaged, 183 821 (99.8%) had received benefits at some point between 1 January 2005 and 30 June 2020 (table 3). Adjusted results indicated that benefit receipt was higher among men, those over the age of 20 years compared with younger patients, those who had subsequently died, had a recorded primary psychiatric diagnosis and patients

living in an area of higher deprivation. Patients from a black ethnic group and those from a mixed ethnic group were more likely to report benefit receipt compared with patients from other ethnic backgrounds.

### Recorded psychiatric diagnosis profile and benefit receipt

Most patients had a primary psychiatric diagnosis recorded in their electronic healthcare record (table 4). About one in five patients (21.6%) were diagnosed with a mood (affective) disorder (eg, depressive episode, mania), followed by disorders due to psychoactive substance abuse (eg, harmful use of drugs or alcohol) (17.5%), and disorders due to physiological conditions (eg, dementia) (17.4%). Benefit receipt across the psychiatric diagnosis spectrum was high, over 80% across most ICD-10 codes, except for behavioural syndromes associated with physiological disturbances and physical factors

**Table 1** Comparison of sociodemographic characteristics of linked and unlinked patients with benefits data (n=239714)

| | Total N (%) | Linked N (%) | Non-linked N (%) | OR (95% CI) for successful linkage | P value | AOR* (95% CI) for successful linkage | P value |
|---|---|---|---|---|---|---|---|
| Overall | 239243 (100.0) | 221243 (100.0) | 18471 (100.0) | – | | | |
| Sex† | 239690 (100.0) | | | | | | |
| Male | | 109321 (49.4) | 8215 (44.5) | Reference | | Reference | |
| Female | | 111921 (50.6) | 10233 (55.5) | 0.82 (0.80 to 0.85)‡ | <0.001 | 0.81 (0.79 to 0.84)‡ | <0.001 |
| Age (years)§ | 239699 (100.0) | | | | | | |
| ≤20 | | 2502 (1.1) | 142 (0.8) | Reference | | Reference | |
| 21–40 | | 77943 (35.2) | 9033 (48.9) | 0.49 (0.41 to 0.58)‡ | <0.001 | 0.49 (0.42 to 0.59)‡ | <0.001 |
| 41–60 | | 75860 (34.3) | 6839 (37.0) | 0.63 (0.53 to 0.75)‡ | <0.001 | 0.62 (0.52 to 0.73)‡ | <0.001 |
| >60 | | 64935 (29.4) | 2445 (13.3) | 1.51 (1.27 to 1.79)‡ | <0.001 | 1.40 (1.17 to 1.66)‡ | <0.001 |
| Ethnicity | 239714 (100.0) | | | | | | |
| White | | 125244 (56.6) | 7405 (40.1) | Reference | | Reference | |
| Black/African/Caribbean/black British | | 30464 (13.8) | 3495 (18.9) | 0.52 (0.49 to 0.54)‡ | <0.001 | 0.56 (0.53 to 0.58)‡ | <0.001 |
| Asian/Asian British | | 10812 (4.9) | 1708 (9.3) | 0.37 (0.35 to 0.40)‡ | <0.001 | 0.40 (0.38 to 0.42)‡ | <0.001 |
| Mixed/multiple racial and ethnic groups | | 4225 (1.9) | 346 (1.9) | 0.72 (0.65 to 0.81)‡ | <0.001 | 0.93 (0.83 to 1.04)‡ | 0.177 |
| Other racial and ethnic minority groups | | 12099 (5.5) | 1889 (10.2) | 0.38 (0.36 to 0.40)‡ | <0.001 | 0.44 (0.42 to 0.46)‡ | <0.001 |
| Not stated¶ | | 38399 (17.4) | 3628 (19.6) | 0.63 (0.60 to 0.65)‡ | <0.001 | 0.74 (0.71 to 0.78)‡ | <0.001 |
| Death† | 239714 (100.0) | | | | | | |
| No | | 174820 (79.0) | 17063 (92.4) | Reference | | Reference | |
| Yes | | 46423 (21.0) | 1408 (7.6) | 3.22 (3.04 to 3.40)‡ | <0.001 | 1.91 (1.79 to 2.03)‡ | <0.001 |
| Deprivation (IMD quintile)†† | 227755 (95.0) | | | | | | |
| First (most deprived) | | 46403 (21.9) | 3390 (21.6) | Reference | | Reference | |
| Second | | 81207 (38.3) | 6536 (41.7) | 0.91 (0.87 to 0.95)‡ | <0.001 | 0.90 (0.86 to 0.94)‡ | <0.001 |
| Third | | 46443 (21.9) | 3546 (22.6) | 0.96 (0.91 to 1.00)‡ | 0.076 | 0.92 (0.87 to 0.96)‡ | 0.001 |
| Fourth | | 23774 (11.2) | 1430 (9.1) | 1.21 (1.14 to 1.29)‡ | <0.001 | 1.09 (1.02 to 1.19)‡ | 0.012 |
| Fifth (least deprived) | | 14165 (6.7) | 779 (5.0) | 1.33 (1.23 to 1.44)‡ | <0.001 | 1.14 (1.05 to 1.24)‡ | 0.001 |
| Resident within local catchment area‡‡ | 227997 (95.0) | | | | | | |
| Yes | | 146860 (69.2) | 11177 (71.2) | 1.06 (1.02 to 1.11)‡ | <0.001 | 1.03 (0.99 to 1.08)‡ | <0.001 |
| No | | 65435 (30.8) | 4525 (28.8) | Reference | | Reference | |
| Number of home/residential postcodes available§§ | 236412 (98.6) | | | | | | |
| 1 | | 118603 (54.2) | 10374 (59.6) | Reference | | Reference | |
| 2 | | 47538 (21.7) | 3474 (20.0) | 1.20 (1.15 to 1.25)‡ | <0.001 | 1.23 (1.19 to 1.29)‡ | <0.001 |
| 3 | | 22252 (10.2) | 1497 (8.6) | 1.30 (1.23 to 1.38)‡ | <0.001 | 1.39 (1.32 to 1.48)‡ | <0.001 |

Continued

Table 1 Continued

| | Total N (%) | Linked N (%) | Non-linked N (%) | OR (95% CI) for successful linkage | P value | AOR* (95% CI) for successful linkage | P value |
|---|---|---|---|---|---|---|---|
| 4 | | 11733 (5.4) | 813 (4.7) | 1.26 (1.17 to 1.36)‡ | <0.001 | 1.41 (1.31 to 1.52)‡ | <0.001 |
| 5 | | 18885 (8.6) | 1243 (7.1) | 1.33 (1.25 to 1.41)‡ | <0.001 | 1.57 (1.47 to 1.67)‡ | <0.001 |

*AOR: adjusted for age (continuous), sex and ethnicity.
†Based on Department for Work and Pensions data, but if missing backfilled with Clinical Records Interactive Search (CRIS) data.
‡P value≤0.01.
§At window end date (30 June 2019), based on CRIS data.
¶Includes not known, not stated or missing.
**Based on CRIS data, but if a death was recorded in benefits data but not recorded in CRIS data it was backfilled accordingly.
††IMD scores published in 2019, postcode used closest and before window end date (30 June 2019).
‡‡Based on Lower-layer Super Output Areas informed by postcode details closest to and before window end date (30 June 2019).
§§Based on five closest postcodes to the earliest accepted referral into SLaM or event within the time window.
AOR, adjusted OR; IMD, Index of Multiple Deprivation.

(56.7%) (eg, eating disorders). Of those receiving benefits, 85.1% received two or more different benefits.

Table 5 provides an overview of selected types of benefits received, namely those related to unemployment, sickness, disability or IS benefits, among patients by recorded primary psychiatric diagnosis code. Most patients diagnosed with a degree of intellectual disabilities (F70–F79) were in receipt of disability benefits such as ESA and DLA or IS benefits such as IS and PIP. These types of benefits were also frequently received by patients diagnosed with pervasive and specific developmental disorders (eg, disturbances in speech and language) (F80–F89)) and patients diagnosed with schizophrenia, schizotypal, delusional disorders and other non-mood psychotic disorders (F20–F29). Unemployment benefit receipt, such as JSA, was most reported among those diagnosed with psychoactive substance abuse (64.0%). Online supplemental table 1 provides an overview of the types of benefits received among the linked patients irrespective of recorded psychiatric diagnosis code, online supplemental table 2 provides an overview of the remaining benefits (eg, RP, PC, AA, WB, BB, ICA, PIB) by recorded primary psychiatric diagnosis code and online supplemental table 3 provides an overview of recorded primary psychiatric diagnosis by UC conditionality type.

## DISCUSSION

We have established an unprecedented data linkage between routinely collected mental healthcare and benefits records, spanning 15 years of linked data, among working-age adults. This enables us to look for the first time, in detail, at the complex longitudinal relationships between mental health and benefit receipt. A linkage rate of 92.3% was achieved using an ad hoc deterministic linkage approach and fuzzy matching. This high linkage rate is comparable to prior data linkages such as CRIS data with Hospital Episode Statistics and Office of National Statistics (ONS) data producing a matching rate of 93.7%,[31] and the CRIS data with the National Pupil Database producing a matching rate of (82.5%).[27]

Despite the high linkage rate, there is still potential for bias, as is often the case when using an ad hoc deterministic approach where no common identifier is available between data sets. Our analysis showed that linkage bias disproportionately affected women, middle aged people and ethnic minority groups. Women may be less likely to be linked because of changes in name and address linked to changes in relationship status, and it has been previously identified that minority groups identifiers are more likely to be entered in error and thus are particularly prone to failure of deterministic linkage processes.[32 33] We also found those with a primary psychiatric diagnosis were more likely to be linked, this may be because of having increased contact with the system and therefore increased opportunity to have personal identifiers recorded that maximise linking opportunity.

**Table 2** Comparison of diagnostic and treatment characteristics of linked and unlinked patients with benefits data (n=239714)

| | Total N (%) | Linked N (%) | Non-linked N (%) | OR (95% CI) for successful linkage | P value | AOR* (95% CI) for successful linkage | P value |
|---|---|---|---|---|---|---|---|
| Overall | 239 714 (100.0) | 221 243 (100.0) | 18 471 (100.0) | – | | | |
| Primary psychiatric diagnosis recorded† | 239 714 (100.0) | | | | | | |
| Yes | | 154 354 (69.8) | 10 997 (59.5) | 1.57 (1.52 to 1.62)‡ | <0.001 | 1.43 (1.38 to 1.48)‡ | <0.001 |
| No | | 66 889 (30.2) | 7474 (40.5) | Reference | | Reference | <0.001 |
| Accessed IAPT§ | 239 714 (100.0) | | | | | | |
| Yes | | 50 899 (23.0) | 3381 (18.3) | 1.33 (1.28 to 1.39)‡ | <0.001 | 1.69 (1.63 to 1.76)‡ | <0.001 |
| No | | 170 344 (77.0) | 15 090 (81.7) | Reference | | Reference | |
| First contact with SLaM | 233 186 (97.3) | | | | | | |
| Before 2013 | | 121 339 (56.4) | 10 989 (61.4) | Reference | | Reference | |
| After 2012 | | 93 936 (43.6) | 6922 (38.7) | 1.23 (1.19 to 1.27)‡ | <0.001 | 1.45 (1.40 to 1.50)‡ | <0.001 |
| Last contact with SLaM | 235 396 (98.4) | | | | | | |
| Before 2013 | | 73 945 (34.0) | 8486 (47.2) | Reference | | Reference | |
| After 2012 | | 143 474 (66.0) | 9491 (52.8) | 1.73 (1.68 to 1.79)‡ | <0.001 | 2.10 (2.04 to 2.19)‡ | <0.001 |
| Age (years) at first presentation to SLaM | 235 204 (98.1) | | | | | | |
| ≤20 | | 23 926 (11.0) | 2106 (11.7) | Reference | | Reference | |
| 21–40 | | 92 178 (42.4) | 10 834 (60.3) | 0.75 (0.71 to 0.79)‡ | <0.001 | 0.67 (0.63 to 0.71)‡ | <0.001 |
| 41–60 | | 55 388 (25.5) | 3593 (20.0) | 1.36 (1.28 to 1.43)‡ | <0.001 | 0.98 (0.90 to 1.07) | 0.637 |
| >60 | | 45 754 (21.1) | 1427 (8.0) | 2.82 (2.63 to 3.02)‡ | <0.001 | 1.53 (1.33 to 1.76)‡ | <0.001 |

*AOR: adjusted for age (continuous), sex and ethnicity.
†Latest psychiatric primary diagnosis recorded closest and before window end date (30 June 2019) based on International Classification of Diseases-10 F codes only (mental and behavioural disorders) but excluding non-specific diagnoses, for example, Z*, F99*, FXX.
‡P value≤0.01.
§Accessed IAPT between 2008 and 30 June 2019.
AOR, adjusted OR; IAPT, Improving Access to Psychological Therapies; SLaM, South London and Maudsley.

**Table 3** Overview of characteristics of those who did or did not ever receive any benefits among linked patients (n=220332)

| | Total N (%) | Never received benefits* N (%) | Ever received benefits N (%) | OR (95% CI) for benefit receipt | P value | AOR† (95% CI) for benefit receipt | P value |
|---|---|---|---|---|---|---|---|
| Overall | 220332 (100.0) | 36511 (100.0) | 183821 (100.0) | – | | – | |
| Sex‡ | 220329 (100.0) | | | | | | |
| Male | | 16550 (45.3) | 92300 (50.2) | Reference | | Reference | |
| Female | | 19961 (54.7) | 91521 (49.8) | 0.82 (0.80 to 0.84)§ | <0.001 | 0.78 (0.77 to 0.80)§ | <0.001 |
| Age (years)¶ | 220329 (100.0) | | | | | | |
| ≤20 | | 1002 (2.7) | 1495 (0.8) | Reference | | Reference | |
| 21–40 | | 18380 (50.3) | 59082 (32.1) | 2.15 (1.99 to 2.34)§ | <0.001 | 2.22 (2.04 to 2.41)§ | <0.001 |
| 41–60 | | 14508 (39.7) | 61028 (33.2) | 2.82 (2.60 to 3.06)§ | <0.001 | 2.79 (2.57 to 3.03)§ | <0.001 |
| >60 | | 2620 (7.2) | 62214 (33.8) | 15.92 (14.56 to 17.40)§ | <0.001 | 15.94 (14.56 to 17.46)§ | <0.001 |
| Ethnicity | 220332 (100.0) | | | | | | |
| White | | 18403 (50.4) | 106251 (57.8) | Reference | | Reference | |
| Black/African/Caribbean/black British | | 2862 (7.8) | 27537 (15.0) | 1.67 (1.60 to 1.74)§ | <0.001 | 1.98 (1.90 to 2.07)§ | <0.001 |
| Asian/Asian British | | 2395 (6.6) | 8387 (4.6) | 0.61 (0.58 to 0.64)§ | <0.001 | 0.67 (0.64 to 0.71)§ | <0.001 |
| Mixed/multiple racial and ethnic groups | | 5879 (1.6) | 3624 (2.0) | 1.07 (0.98 to 1.17)§ | 0.138 | 1.73 (1.58 to 1.89)* | <0.001 |
| Other racial and ethnic minority groups | | 2850 (7.8) | 9204 (5.0) | 0.56 (0.53 to 0.59)§ | <0.001 | 0.72 (0.68 to 0.75)§ | <0.001 |
| Not stated** | | 9414 (25.8) | 28818 (15.7) | 0.53 (0.52 to 0.55)§ | <0.001 | 0.72 (0.70 to 0.74)§ | <0.001 |
| Death‡ | 220332 (100.0) | | | | | | |
| No | | 34935 (95.7) | 139017 (75.6) | Reference | | Reference | |
| Yes | | 1576 (4.3) | 44804 (24.4) | 7.14 (6.79 to 7.52)§ | <0.001 | 2.77 (2.61 to 2.93)§ | <0.001 |
| Deprivation (IMD quintile)†† | 211276 (95.9) | | | | | | |
| First (most deprived) | | 4956 (14.2) | 41296 (23.4) | Reference | | Reference | |
| Second | | 12323 (35.3) | 68580 (38.9) | 0.67 (0.64 to 0.69)§ | <0.001 | 0.64 (0.61 to 0.66)§ | <0.001 |
| Third | | 9013 (25.8) | 37264 (21.1) | 0.50 (0.48 to 0.52)§ | <0.001 | 0.49 (0.47 to 0.50)§ | <0.001 |
| Fourth | | 5266 (15.1) | 18442 (10.5) | 0.42 (0.40 to 0.44)§ | <0.001 | 0.41 (0.39 to 0.43)§ | <0.001 |
| Fifth (least deprived) | | 3404 (9.7) | 10732 (6.1) | 0.38 (0.36 to 0.40)§ | <0.001 | 0.37 (0.35 to 0.39)§ | <0.001 |
| Primary psychiatric diagnosis recorded‡‡ | 220332 (100.0) | | | | | | |
| Yes | | 22060 (60.4) | 131702 (71.7) | 1.66 (1.62 to 1.69)§ | <0.001 | 1.29 (1.26 to 1.33)§ | <0.001 |
| No | | 14451 (39.6) | 52119 (28.4) | Reference | | Reference | |

Continued

**Table 3** Continued

| | Total N (%) | Never received benefits* N (%) | Ever received benefits N (%) | OR (95% CI) for benefit receipt | P value | AOR† (95% CI) for benefit receipt | P value |
|---|---|---|---|---|---|---|---|
| Accessed IAPT§§ | 220332 (100.0) | | | | | | |
| Yes | | 9707 (26.6) | 41003 (22.3) | 0.79 (0.77 to 0.81)§ | | 1.01 (0.99 to 1.04) | |
| No | | 26804 (73.4) | 142818 (77.7) | Reference | <0.001 | Reference | 0.284 |

*This includes patients who did not have a benefits record entry as well as those who did have an entry but did not receive any benefits.
†AOR: adjusted for age (continuous), sex and ethnicity.
‡Based on Department for Work and Pensions data, but if missing backfilled with Clinical Records Interactive Search (CRIS) data.
§P value≤0.01.
¶At window end date (30 June 2019), based on CRIS data.
**Includes not known, not stated or missing.
††IMD scores published in 2019, postcode used closest and before window end date (30 June 2019).
‡‡Latest psychiatric primary diagnosis recorded closest and before window end date (30 June 2019) based on International Classification of Diseases-10 F codes only (mental and behavioural disorders) but excluding non-specific diagnoses, for example, Z*, F99*, FXX.
§§Accessed IAPT between 2008 and 30 June 2019.
AOR, adjusted OR; IAPT, Improving Access to Psychological Therapies; IMD, Index of Multiple Deprivation.

Of patients accessing SLaM services and successfully linked, 83% had engaged with the DWP, and of those, 99.8% had received a benefit of any kind. This finding is not unexpected and are in accordance with previous research showing that one of the most reported working-age disabilities and reason for claiming unemployment and sickness-related benefits is a mental health problem.[1] We found those who were men, over 20 years old, had subsequently died, had a primary psychiatric diagnosis, were of a black ethnic group or mixed ethnic group and lived in a higher area of deprivation were all more likely to have received a benefit. Most received benefits among the sample included ESA, JSA and IS.

Further, of those who received UC (n=46 789), a high proportion was placed in the UC conditionality regime—searching for work group (n=38 073, 81.4%). Next, we can explore what support and work adjustments this group are able to access in relation to finding work. We also showed that over half of the sample had received a psychiatric diagnosis, with one in five having been diagnosed with a mood affective disorder. It is likely that those with a psychiatric diagnosis are more likely to fall out of work and therefore more likely to claim sickness and unemployment related benefits. A comparison of levels of benefit receipt and patterns among the UK working age population is out of scope for this paper but will be explored in detail in the future. However, we know that, for example, approximately 9.9 million working-age people were claiming a combination of benefits in 2021, including UC, PIP, DLA, HB, AA, ESA, JSA, and IS.[34 35]

Previous population-based research reporting on mental health and benefit receipt in the UK has been limited in its use of self-report survey data, as well as a basic level of detail in relation to benefit receipt. For example, the Adult Psychiatric Morbidity Survey (APMS) (2014) showed that a large proportion of people receiving ESA reported symptoms of a mental disorder, supporting our initial findings. Nevertheless, the APMS did not have data on newer benefits (eg, UC) and were unable to distinguish between the level of benefit and payment received within a particular benefit type or provide other important data such as details of the WCA process.[36] Our findings are also comparable to other studies that show a large proportion of people who receive benefits report symptoms of a mental disorder.[6 7] Finally, ONS holds data reflecting labour market activity and collects information via the Labour Force Survey relating to (un)employment, counts of benefit claimants and selected self-reported physical and mental health conditions. However, detailed, longitudinal health data is not available.[37]

Though we are yet to explore sickness and disability related benefits among our sample in detail, much research into disability pension (DP) awards has already been conducted in Norway using large population-based cohorts containing mental and physical health data linked to national databases of disability benefits using national identification numbers. For example, one study investigated the impact of anxiety and depression on DPs

**Table 4** Overview of recorded primary psychiatric diagnoses in linked patients (n=153 762) and whether patients who were given a diagnosis had received benefits (n=131 702)

| | Recorded primary psychiatric diagnoses* (ICD-10 code and description) N (%) | Received a benefit† N (%) |
|---|---|---|
| F00–F09 (mental and behavioural disorders, and mental disorders due to known physiological conditions) | 26 775 (17.4) | 26 069 (97.4) |
| F10–F19 (mental and behavioural disorders due to psychoactive substance use) | 26 879 (17.5) | 23 731 (88.2) |
| F20–F29 (schizophrenia, schizotypal, delusional disorders and other non-mood psychotic disorders) | 16 082 (10.5) | 14 944 (92.9) |
| F30–F39 (mood (affective) disorders) | 33 235 (21.6) | 27 046 (81.4) |
| F40–F48 (anxiety, dissociative, stress-related, somatoform and other non-psychotic mental disorders) | 25 944 (16.9) | 20 432 (78.8) |
| F50–F59 (behavioural syndromes associated with physiological disturbances and physical factors) | 6773 (4.4) | 3840 (56.7) |
| F60–F69 (disorders of adult personality and behaviour) | 6219 (4.0) | 5495 (88.4) |
| F70–F79 (intellectual disabilities) | 2484 (1.6) | 2448 (98.6) |
| F80–F89 (pervasive and specific developmental disorders) | 2904 (1.9) | 2623 (90.3) |
| F90–F98 (behavioural and emotional disorders with onset usually occurring in childhood and adolescence) | 6467 (4.2) | 5092 (78.7) |

*Latest psychiatric primary diagnosis recorded closest and before window end date (30 June 2019) based on ICD-10 F codes only (mental and behavioural disorders) but excluding non-specific diagnoses, for example, Z*, F99*, FXX.
†Any type of benefits received between 1 January 2005 and 30 June 2020. % will not add up to 100% as patients could have received multiple benefits over time.
ICD, International Classification of Diseases.

awarded for mental health and physical health diagnoses. They showed long-term occupational impact of anxiety and depression symptoms and their subsequent independent contribution towards DPs awarded.[23] Another study linking mental health cohort data and the National Insurance Administration database containing DP award data showed that anxiety and depression at baseline were strongly associated with receiving a DP award at follow-up.[22] A Finnish study found that there was evidence of regional variation in mental disorder DP and mental health service factors, a critical finding when considering service provision.[21]

There is great volume and depth of data available in the newly established linkage. Clinical data from SLaM provides detail on both primary and secondary diagnoses, in addition to diagnosis severity as measured using the Health of the Nation Outcome Scale, and data on appointment history and clinical intervention provision. As SLaM is one of the largest secondary mental healthcare services in the UK, findings may be generalisable to other settings, though considerations of key differences at local level, for example type of mental healthcare services provided and the profile of patients accessing services in a highly populated, ethnically diverse urban area, should be given. In addition, SLaM provides a variety of national and specialist services, such as a specialist affective disorders service, meaning that some patients will be residing outside the SLaM catchment area. Benefits data provides extensive detail on number, type and amounts of benefits received, as well as data on interventions accessed and the WCA process. Further, the longitudinal nature of the data helps to ensure that those who engage intermittently with the welfare or mental healthcare system can still be captured where this would be more challenging in cross-sectional research or studies spanning a shorter period.

However, there are limitations of the linked data. For example, due to prior legalities, our sample includes only those who have been referred to SLaM, meaning we cannot directly compare our findings to those who have not accessed secondary mental healthcare services, but may have received benefits. In addition, as neither data set holds well populated or accurate employment related data, a proxy for returning to work is considered where someone is no longer receiving an unemployment related benefit. However, there can be varying reasons as to why someone stops receiving this type of benefit, other than because they have found work, such as no longer meeting the eligibility criteria or having a benefit suspended because of a sanction. The lack of this information may disproportionally impact vulnerable groups who are likely to have disengaged with the benefits system, such as homeless people or refugees, and still not have found work or be consistently in work. It should also be noted that interpretation of findings should consider the level of uptake and possible benefit underclaiming in the current sample.[38] Notwithstanding this, the data we hold for UC, but not for other unemployment related legacy benefits provides information that indicates whether someone is

**Table 5** Overview of patients who had a recorded primary psychiatric diagnosis and benefit receipt related to unemployment, sickness, disability, income support benefits

| Benefit type* Recorded primary psychiatric diagnoses (ICD-10 code and description) † | Universal Credit N (%) n=30622 | Jobseeker's Allowance N (%) n=50076 | Employment and Support Allowance N (%) n=60681 | Incapacity Benefit N (%) n=38336 | Severe Disability Allowance N (%) n=2957 | Personal Independence Payment N (%) n=35214 | Disability Living Allowance N (%) n=40189 | Income Support N (%) n=43451 |
|---|---|---|---|---|---|---|---|---|
| F00–F09 (mental and behavioural disorders and mental disorders due to known physiological conditions) n=26069 | 513 (2.0) | 1352 (5.2) | 2074 (8.0) | 2333 (9.0) | 178 (0.7) | 1600 (6.1) | 3734 (14.3) | 1606 (6.2) |
| F10–F19 (mental and behavioural disorders due to psychoactive substance use) n=23713 | 8574 (36.2) | 15167 (64.0) | 15563 (65.6) | 10683 (45.1) | 171 (0.7) | 6165 (26.0) | 4811 (20.3) | 11334 (47.8) |
| F20–F29 (schizophrenia, schizotypal, delusional disorders and other non-mood psychotic disorders) n=14944 | 2898 (19.4) | 4865 (32.6) | 9757 (65.3) | 7176 (48.0) | 975 (6.5) | 6166 (41.3) | 8662 (58.0) | 7202 (48.2) |
| F30–F39 (mood (affective) disorders) n=27046 | 7044 (26.0) | 11351 (42.0) | 12486 (46.2) | 8076 (29.9) | 360 (1.3) | 7232 (26.7) | 7840 (29.0) | 9621 (35.6) |
| F40–F48 (anxiety, dissociative, stress-related, somatoform and other non-psychotic mental disorders) n=20432 | 5451 (26.7) | 8612 (42.2) | 9743 (47.7) | 5543 (27.1) | 199 (1.0) | 6097 (29.8) | 5804 (28.4) | 6661 (32.6) |
| F50–F59 (behavioural syndromes associated with physiological disturbances and physical factors) n=3840 | 1168 (30.4) | 2124 (55.3) | 1406 (36.6) | 685 (17.8) | 24 (0.6) | 824 (21.5) | 810 (21.1) | 1027 (26.7) |
| F60–F69 (disorders of adult personality and behaviour) n=5495 | 1874 (34.1) | 2640 (48.0) | 3820 (69.5) | 2095 (38.1) | 114 (2.1) | 2615 (47.6) | 2256 (41.1) | 2722 (49.5) |
| F70–F79 (intellectual disabilities) n=2448 | 238 (9.7) | 246 (10.1) | 1856 (75.7) | 637 (26.0) | 848 (34.6) | 1330 (54.3) | 2255 (92.1) | 1451 (59.3) |
| F80–F89 (pervasive and specific developmental disorders) n=2623 | 653 (24.9) | 900 (34.3) | 1598 (60.9) | 448 (17.1) | 66 (2.5) | 1447 (55.2) | 1711 (65.2) | 558 (21.3) |
| F90–F98 (behavioural and emotional disorders with onset usually occurring in childhood and adolescence) n=5092 | 2209 (43.4) | 2819 (55.4) | 2378 (46.7) | 660 (13.0) | 22 (0.4) | 1738 (34.1) | 2306 (45.3) | 1269 (24.9) |

*Any type of benefits received between 1 January 2005 and 30 June 2020.
†Latest psychiatric primary diagnosis recorded closest and before window end date (30 June 2019) based on ICD-10 F codes only (mental and behavioural disorders) but excluding non-specific diagnoses, for example, Z*, F99*, FXX.
ICD, International Classification of Diseases.

in or out of work. Future projects should consider the important advantages of further linking employment related data, held by Her Majesty's Revenue and Customs in the UK, to the current linked data, as well as including a case–control population comparison group who were not referred to SLaM services.

Despite the limitations, this novel data linkage between routinely collected electronic mental healthcare records and benefits records contains extensive time-variant data that allows us to explore the bidirectional and complex relationships between mental health, employment and benefit receipt, something that has not previously been seen in the UK. It provides opportunity for retrospective longitudinal cohort studies to be carried out and provide understanding of how best to design and provide the most effectively tailored interventions to target different patient groups and benefit claimants. So far, we have shown that a very high percentage of those in contact with secondary mental healthcare services have received a benefit at some point within the 15-year window our linked data spans. We can now look in further detail at this population to answer important research questions and address areas of interest such as the impact of UC and WCA on people with mental disorders, the effectiveness of certain interventions to support people to return to work and the general trends and trajectories of benefit receipt among people accessing secondary mental healthcare services. High-quality outputs can be produced providing much needed evidence relating to both occupational and welfare policy initiatives and interventions within the joint DWP/Department of Health and Social Care Work and Health Unit, and NHS mental healthcare providers, all with the aim of improving outcomes for people with mental health problems.

**Author affiliations**
[1]Department of Psychological Medicine, King's College London, Institute of Psychiatry, Psychology and Neuroscience, London, UK
[2]King's Centre for Military Health Research, Department of Psychological Medicine, King's College London, Institute of Psychiatry, Psychology and Neuroscience, London, UK
[3]South London and Maudsley Mental Health NHS Trust, NIHR Maudsley Biomedical Research Centre, London, UK
[4]Population Health Sciences, Bristol Medical School, University of Bristol, Bristol, UK
[5]Centre for Implementation Science, Health Services and Population Research Department, King's College London, Institute of Psychiatry, Psychology and Neuroscience, London, UK
[6]Department of Biostatistics and Health Informatics, King's College London, Institute of Psychiatry, Psychology and Neuroscience, London, UK
[7]Department of Occupational Health, Guy's and St Thomas' Hospitals NHS Trust, London, UK
[8]King's College London, London, UK
[9]Academic Department of Military Mental Health, Department of Psychological Medicine, King's College London, Institute of Psychiatry, Psychology and Neuroscience, London, UK

**Acknowledgements** We would like to thank Megan Pritchard at the National Institute for Health and Care Research (NIHR) Maudsley Biomedical Research Centre for their support with this study. We would also like to thank the members of the NIHR Maudsley Biomedical Research Nucleus Data Linkage Service User and Carer Advisory Group for their input. We are very grateful to the DWP/Department of Health and Social Care joint Work and Health Unit staff, especially staff working in the Joint Health and Work Unit, who supported us in creating this linked dataset and advice provided.

**Contributors** SS conceptualised and designed the study with input from AP, AB, SD, IB, NTF, MH, IM and JD. MB, RL and AJ took the lead in data curation. SS and AP led on the methodology, formal analysis and project administration. MB, JD, SD, RL, IB and AJ supported the methodology. SS acquired funding for the study with support from NTF, IM and MH. Supervision was provided by NTF, MH and IM. SS wrote the initial draft of this paper (Introduction, Methods, Results). AP wrote the initial draft of the discussion. SS and AP revised the paper. All authors commented on the final draft of this paper. SS had final responsibility for the decision to submit for publication.

**Funding** This paper represents independent research funded by the National Institute for Health and Care Research (NIHR), as part of the corresponding author's NIHR Advanced Fellowship (ref: NIHR 300592). This paper represents independent research part funded by the NIHR Maudsley Biomedical Research Centre at South London and Maudsley NHS Foundation Trust and King's College London (ref: N/A). MH is an NIHR senior investigator. The views expressed are those of the authors and not necessarily of the NIHR, the Department of Health and Social Care or the Department for Work and Pensions. The funder had no contribution in the study design, data collection, analysis and interpretation of the data, manuscript writing and the decision to submit the paper for publications.

**Competing interests** MH is principal investigator of RADAR-CNS consortium—a public private partnership in collaboration with five pharmaceutical companies—Janssen, Biogen, UCB, MSD and Lundbeck, outside of the submitted work.

**Patient and public involvement** Patients and/or the public were involved in the design, or conduct, or reporting, or dissemination plans of this research. Refer to the Methods section for further details.

**Patient consent for publication** Not applicable.

**Ethics approval** We submitted the proposed linkage to the South Central—Oxford C Research Ethics Committee for ethical approval. A favourable opinion was received in 2017 (ref 17/SC/0581). In addition, we successfully applied in 2017 for Section 251 approval under the NHS Health Research Authority Confidential Advisory Group (ref 17CAG0055). We believed that it was not practical or appropriate for the proposed linkage to be successfully achieved through a consent-based methodology. Once ethical approvals were in place, we developed a data sharing agreement. This agreement outlines the data sharing agreements between SLaM and the Department for Work and Pensions in relation to the data linkage. The agreement sets out lawful basis of the data linkage as well as the principles and procedures for data sharing and the use of the linked data.

**Provenance and peer review** Not commissioned; externally peer reviewed.

**Data availability statement** Data may be obtained from a third party and are not publicly available. Data are not publicly available. Access to deidentified data can be applied for via the NIHR Maudsley Biomedical Research Centre at the South London and Maudsley NHS Foundation Trust, upon reasonable request. Requests for data will be considered on a case-by-case basis, given the sensitive nature of the data, and access will only be granted if approval is given by the Work and Health Screening Panel and other governance requirements are fulfilled. For more information, please contact: cris.administrator@slam.nhs.uk.

**ORCID iDs**
Sharon A M Stevelink http://orcid.org/0000-0002-7655-7986
Sarah Dorrington http://orcid.org/0000-0002-6462-1880
Amelia Jewell http://orcid.org/0000-0002-0887-2159

Nicola T Fear http://orcid.org/0000-0002-5792-2925
Johnny Downs http://orcid.org/0000-0002-8061-295X

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
