## [Reviewer comments · BMJ Open]

ARTICLE DETAILS

TITLE (PROVISIONAL)	Linking electronic mental healthcare and benefits records in South London: design, procedure, and descriptive outcomes.
AUTHORS	Stevelink, Sharon; Phillips, Ava; Broadbent, Matthew; Boyd, Andy; Dorrington, Sarah; Jewell, Amelia; Leal, Ray; Bakolis, Ioannis; MADAN, IRA; Hotopf, Matthew; Fear, Nicola; Downs, Johnny

VERSION 1 – REVIEW

REVIEWER	Brown, Heather Lancaster University, Division of Health Research
REVIEW RETURNED	24-Aug-2022

GENERAL COMMENTS	Abstract: As per my comments below and in the discussion, I would clarify that this is a first for data linkage in the UK but it is not the first of this type of data linkage as this is already done in Nordic countries. Introduction: As I also mention in the discussion section, it may be worthwhile to put the research into an international context by mentioning some of the linkage of health and benefit data in Nordic countries and how it is used for service planning. Methods: Why did you choose a fuzzy matching? As a robustness check did you try any other matching techniques? I also think it would be helpful to include potentially as a supplemental file, a brief description of the different types of benefits so the reader can get a better idea of the context for the results in Tables 4 and 5. Results: Overall, I think the results section would benefit from a re-write. It is hard to follow the results and see how all the Tables fit together. It may be helpful to combine Tables 4 and 5 as there is not really much description of Table 4 in the text. On page 12, the description around age and linkage is confusing to follow. So very young adults <21 were more likely to be linked than middle aged adults but older adults were more likely to be linked than both middle aged and younger adults (which I can see is the case from the table, but is not really described in a clear way in the text) Discussion: In the discussion it would be worthwhile discussing how the findings relate to some of the Nordic literature from countries such as Finland
---

	where such linkage is possible because of a universal identifier and is used for policy making. See for example: Karolaakso T, Autio R, Näppilä T, Leppänen H, Rissanen P, Tuomisto MT, Karvonen S, Pirkola S. Contextual and mental health service factors in mental disorder-based disability pensioning in Finland—a regional comparison. BMC health services research. 2021 Dec;21(1):1-3.
--	--

REVIEWER	Tseliou, Foteini Cardiff University
REVIEW RETURNED	03-Oct-2022

GENERAL COMMENTS	This is a well-conducted study that undertook a novel data-linkage of routine records from UK health and welfare government service providers within a secure research environment. However, I have some comments on the methods implemented and how these could affect the interpretation of the observed results.  • Were there any inconsistencies or issues when trying to collate information in the format of free clinical text notes with those within structured fields? Knowing that the level of information provided in records can vary significantly, this could be an issue. • Is women's low linkage rate linked to changing name due to marriage? Could that be accounted for via marriage status? • It is mentioned that a higher number of historical postcodes was associated with more successful linkage. Could this be linked to the fact that only up to five addresses could be included? Could this indicate that individuals who have moved more than 5 times during the study period might be missed out? • In terms of the lack of comparison group, could this be resolved with the use of healthcare records on physical conditions? • If there were multiple diagnoses across years was one of them chosen as the primary diagnosis? How was that determined? For example, if one individual had a primary long-term physical condition e.g. heart condition and a secondary mental health condition e.g. depression, would they be included? • Were there any individuals that had data in both SLAM services and IAPT services? Was it possible to control for that so that there is no overlap/ duplication? This was not very clear. • Has there been any feedback by the Advisory group on the preliminary findings presented in this paper e.g. around the choice of a sample with more severe mental health symptomatology? If yes, it should be mentioned in the results or discussion section. • Why were individuals who had resided in Northern Ireland excluded? Was that decided on the basis on a different policy on benefits receipt or receipt of treatment? • It would be interesting to further explore how individuals who engaged with DWP but were not successful in claiming benefits differed from those who successfully claimed benefits especially in terms of their mental health. • Why are the "contact with SLAM" variables categorized into before 2010 and after 2010? Did a policy change take place on that year? • In Table 4, is there an overlap between different types of benefits received e.g. being in receipt of ICA and DLA? This should be acknowledged in text or perhaps create a composite variable for multiple benefits receipt as these individuals would be different from those who only receive one type of benefits.
--

VERSION 1 – AUTHOR RESPONSE

Reviewer 1	Response
Abstract: As per my comments below and in the discussion, I would clarify that this is a first for data linkage in the UK but it is not the first of this type of data linkage as this is already done in Nordic countries.	We have clarified this throughout the manuscript as follows: This data linkage is the first of its kind in the UK to demonstrate the use of routinely collected mental health and benefits data (Abstract, page 2). There are no pre-existing datasets in the UK that can currently address this (Introduction, page 4). Previous population-based research reporting on mental health and benefit receipt in the UK has been limited in its use of self-report survey data, as well as a basic level of detail in relation to benefit receipt (Discussion, page 21). Despite the limitations, this novel data linkage between routinely collected electronic mental healthcare records and benefits records contains extensive time-variant data that allows us to explore the bidirectional and complex relationships between mental health, employment and benefit receipt, something that has not previously been seen in the UK (Discussion, page 22).
Introduction: As I also mention in the discussion section, it may be worthwhile to put the research into an international context by mentioning some of the linkage of health and benefit data in Nordic countries and how it is used for service planning.	We have incorporated the following in the Introduction as suggested: However, research into welfare and mental health using data registries have been led by those in Nordic countries where a unique personal identifier is available to all those with a permanent residence record, paving the way for opportunities in linkages between health and welfare registers (21). (Introduction, page 4).
Methods: Why did you choose a fuzzy matching? As a robustness check did you try any other matching techniques?	The matching process was led by the Department for Work and Pensions and their standard matching process only uses deterministic matching for administrative data. An ad hoc fuzzy matching deterministic technique

	was used to allow for agreement on partial identifiers. As the Department for Work and Pensions led on the matching, we were not able to implement any other matching techniques. However, the high uniqueness cut-off threshold applied for the fuzzy matching, and the resulting high linkage rate support the robustness of the deterministic matching process undertaken.
I also think it would be helpful to include potentially as a supplemental file, a brief description of the different types of benefits so the reader can get a better idea of the context for the results in Tables 4 and 5.	We have referenced a link to the UK Government website providing an overview of the benefits in the Methods section of the manuscript (https://www.gov.uk/browse/benefits) As the eligibility criteria for benefits are subject to change, as well as how much people are entitled to, we believe this information will remain most accurate to readers by including this link, instead of providing a supplementary file. A most recent example is the increase in the number of hours people need to work on a weekly basis to retain their full entitlement to Universal Credit.
In the discussion it would be worthwhile discussing how the findings relate to some of the Nordic literature from countries such as Finland where such linkage is possible because of a universal identifier and is used for policy making. See for example: Karolaakso T, Autio R, Näppilä T, Leppänen H, Rissanen P, Tuomisto MT, Karvonen S, Pirkola S. Contextual and mental health service factors in mental disorder-based disability pensioning in Finland—a regional comparison. BMC health services research. 2021 Dec;21(1):1-3.	We have incorporated the following in the Discussion as suggested: Though we are yet to explore sickness and disability related benefits among our sample in detail, much research into disability pension (DP) awards has already been conducted in Norway using large population-based cohorts containing mental and physical health data linked to national databases of disability benefits using national identification numbers. For example, one study investigated the impact of anxiety and depression on DPs awarded for mental health and physical health diagnoses. They showed long-term occupational impact of anxiety and depression symptoms and their subsequent independent contribution towards DPs

	awarded (23). Another study linking mental health cohort data and the National Insurance Administration database containing DP award data showed that anxiety and depression at baseline were strongly associated with receiving a DP award at follow-up (22). A Finnish study found that there was evidence of regional variation in mental disorder DP and mental health service factors, a critical finding when considering service provision (21). (Discussion, page 22).
Overall, I think the results section would benefit from a re-write. It is hard to follow the results and see how all the Tables fit together. It may be helpful to combine Tables 4 and 5 as there is not really much description of Table 4 in the text.	We have moved Table 4 to the supplementary materials to improve the flow of the results. We have also carefully re-read the Results section and made several amendments to improve readability.
On page 12, the description around age and linkage is confusing to follow. So very young adults <21 were more likely to be linked than middle aged adults but older adults were more likely to be linked than both middle aged and younger adults (which I can see is the case from the table, but is not really described in a clear way in the text),	Thank you for highlighting this. We now write: Results from adjusted logistic regression analyses indicated that the following groups of patients were less likely to be linked (an OR greater than 1 denotes greater chance of successful linkage compared with the reference): female patients vs. male patients, ethnic minority groups vs. patients from a white ethnic background, and middle-aged patients vs. younger patients (<21 years) and patients with only one postcode available vs. two or more postcodes. Compared to younger patients (<21 years), middle-aged patients (21-60 years) were less likely to be successfully linked, whereas older patients (>60 years) were more likely to be linked compared to all other age groups (Table 1). Further, the linkage rate was also higher among patients who had a higher number of historical postcodes available. On the other hand, older patients (>60 years) were more likely to be linked than younger patients. (Results, page 10).
Reviewer 2	
This is a well-conducted study that undertook a novel data-linkage of routine records from UK health and welfare government service providers within a secure research environment. However, I have some comments on the	Thank you.

methods implemented and how these could affect the interpretation of the observed results.	
Were there any inconsistencies or issues when trying to collate information in the format of free clinical text notes with those within structured fields? Knowing that the level of information provided in records can vary significantly, this could be an issue.	For the current data linkage paper, no data was used from the clinical text notes. Only data from structured fields as part of the patient's electronic mental healthcare record was used. We agree that this is something we need to explore carefully in future papers. We now write: For the current paper, only data from structured fields were used. (Methods, page 5).
Is women's low linkage rate linked to changing name due to marriage? Could that be accounted for via marriage status?	As outlined in the Discussion of the manuscript, the lower linkage rate is indeed likely to be due to changes in relationship status (see page 21). Unfortunately, marital status is not consistently recorded in the electronic mental healthcare records (and absent in benefits records).
It is mentioned that a higher number of historical postcodes was associated with more successful linkage. Could this be linked to the fact that only up to five addresses could be included? Could this indicate that individuals who have moved more than 5 times during the study period might be missed out?	If patients had more than five historical postcodes, they were still included in the sample that was sent for linkage to the Department for Work and Pensions. In this case, only the five postcodes closest to when a referral to SLaM was accepted or when an event was recorded were included. This is now detailed under Table 1: Number of home/residential postcodes available-: -based on five closest postcodes to the earliest accepted referral into SLaM or event within the time window (Table 1, page 12).
In terms of the lack of comparison group, could this be resolved with the use of healthcare records on physical conditions?	Welfare records are only linked with the electronic mental healthcare records, and this was only achieved after approximately 5 years of negotiations with the Department for Work and Pensions. We are keen to extend the linkage of welfare records with other data sources, however this is not a current priority. We want to capitalise first on this novel data linkage.
If there were multiple diagnoses across years was one of them chosen as the primary diagnosis? How was that determined? For example, if one individual had a primary long-term physical condition e.g. heart condition and a secondary mental health	As outlined in the Materials section of the Methods, the psychiatric primary diagnosis recorded in a patient's

condition e.g. depression, would they be included?	record closest and before the SLaM window end date was taken. We have included this description under the Tables where appropriate. If patients received another diagnosis at the same date, this would have been recorded by the clinician as a psychiatric secondary diagnosis. It is unlikely that patients were diagnosed with a long-term physical condition as SLaM is a mental health service provider. Even if that was the case, the secondary mental health condition would still be included as we specified our interest in psychiatric diagnoses during the clinical data extraction phase.
Were there any individuals that had data in both SLaM services and IAPT services? Was it possible to control for that so that there is no overlap/ duplication? This was not very clear.	Some patients accessed both SLaM specialist services and IAPT. However, data are recorded in two different electronic mental health systems, namely ePJS (SLaM) and IAPTus (IAPT). Therefore, there was no duplication or overlap in the data used for this data linkage paper as only data extracted from ePJS was used.
Has there been any feedback by the Advisory group on the preliminary findings presented in this paper e.g. around the choice of a sample with more severe mental health symptomatology? If yes, it should be mentioned in the results or discussion section.	The Advisory Group has been informed of the findings presented in this paper via a newsletter. No feedback was received.
Why were individuals who had resided in Northern Ireland excluded? Was that decided on the basis on a different policy on benefits receipt or receipt of treatment?	The Health Research Authority approval under Regulation 5 of the Health Service (Control of Patient Information) Regulations 2002 to process patient identifiable information without consent only applies to England and Wales. In addition, the devolved legislature of Northern Ireland is responsible for administering benefits to patients who resided in Northern Ireland at the time of their benefit receipt. Therefore, the DWP is not allowed to share this data. We did not have the approvals to maintain this small group of patients (n=795 out of 221,243 unique records) in our linked dataset. We now added the as a footnote (Results, page 9).
It would be interesting to further explore how individuals who	This is an interesting avenue

engaged with DWP but were not successful in claiming benefits differed from those who successfully claimed benefits especially in terms of their mental health.	for further research and something we hope to explore in the future.
Why are the “contact with SLaM” variables categorized into before 2010 and after 2010? Did a policy change take place on that year?	We have now changed this to 2013 as in that year Personal Independence Payment was introduced, thereby replacing Disability Living Allowance. We now write: This cut off was chosen as Personal Independence Payment was introduced in 2013 (Methods, page 8). We have also updated this in Table 2 accordingly (Table 2, page 13).
In Table 4, is there an overlap between different types of benefits received e.g. being in receipt of ICA and DLA? This should be acknowledged in text or perhaps create a composite variable for multiple benefits receipt as these individuals would be different from those who only receive one type of benefits.	We have moved Table 4 to the supplementary material based on feedback from Reviewer 1. Some patients received different benefits over the 15-year time period. We now write: Of those receiving benefits, 85.1% received 2 or more different benefits. (Results, page 17). Exploration of multiple benefit receipt is out of scope for the current paper.
Editor comments	
Please revise the ‘Strengths and limitations of this study’ section of your manuscript (after the abstract). This section should contain up to five short bullet points, no longer than one sentence each, that relate specifically to the methods. The novelty, aims, results or expected impact of the study should not be summarised here.	We have shortened the bullet points as requested. We now write:  • This is a novel data linkage between electronic mental healthcare records and benefits records in the UK, providing the opportunity to answer important questions relating to mental health, work and benefit receipt. • A strength of this data linkage is the a high linkage rate of 92.3%. was achieved. • The sample does not include a comparison group (e.g., people who did not access

	secondary mental healthcare services).  • Although there are indicators of people being in and out of work depending on what type of benefits are being received (unemployment related benefits). There is no reliable employment variable within the data stating whether someone is currently in or out of work (except for Universal Credit). • There is a potential for linkage bias as a result of the method used (ad hoc deterministic fuzzy matching) and having no unique identifier between data sets (Strengths and limitations section, page 3). We have carefully proofread the revised manuscript and made a few amendments accordingly. These are highlighted in Track Changes for clarity.
--	--

VERSION 2 – REVIEW

REVIEWER	Brown, Heather Lancaster University, Division of Health Research
REVIEW RETURNED	01-Dec-2022
GENERAL COMMENTS	Thank you for addressing my comments.